# Flame Retardant Behavior of Ternary Synergistic Systems in Rigid Polyurethane Foams

**DOI:** 10.3390/polym11020207

**Published:** 2019-01-24

**Authors:** Wang Xi, Lijun Qian, Linjie Li

**Affiliations:** 1Shandong Key Laboratory of Marine Fine Chemicals, Shandong Ocean Chemical Industry Scientific Research Institute, Weifang 262737, China; xwbtbu@126.com; 2School of Materials Science & Mechanical Engineering, Beijing Technology and Business University, Beijing 100048, China; lljbtbu@163.com; 3Engineering Laboratory of Non-halogen Flame Retardants for Polymers, Beijing 100048, China

**Keywords:** flame retardancy, foams, phosphorus, ternary synergistic effect

## Abstract

In order to explore flame retardant systems with higher efficiency in rigid polyurethane foams (RPUFs), aluminum hydroxide (ATH), [bis(2-hydroxyethyl)amino]-methyl-phosphonic acid dimethyl ester (BH) and expandable graphite (EG) were employed in RPUF for constructing ternary synergistic flame retardant systems. Compared with binary BH/EG systems and aluminum oxide (AO)/BH/EG, ATH/BH/EG with the same fractions in RPUFs demonstrated an increase in the limited oxygen index value, a decreased peak value of heat release rate, and a decreased mass loss rate. In particular, it inhibited smoke release. During combustion, ATH in ternary systems decomposed and released water, which captured the phosphorus-containing products from pyrolyzed BH to generate polyphosphate. The polyphosphate combined with AO from ATH and the expanded char layer from EG, forming a char layer with a better barrier effect. In ternary systems, ATH, BH, and EG can work together to generate an excellent condensed-phase synergistic flame retardant effect.

## 1. Introduction

Rigid polyurethane foams (RPUFs) are widely used in thermal insulation, space filling, and other applications due to their excellent properties, which include excellent low heat conductivity, light weight, high compressive strength, low moisture permeability, and electrical insulating properties [1,2,3,4]. However, RPUF is also an easily flammable material, but most of its applications have the requirement of flame retardancy [5,6,7]. Thus, if the flammability of RPUFs were not improved, RPUFs would be limited in their application range due to the absence of anti-fire safety. Therefore, different addition-type and reactive-type flame retardants have been employed to prepare RPUF matrices with flame retardancy [8,9,10].

These flame retardants are usually based on certain elements, such as phosphorus, nitrogen, or halogens [11,12,13]. The reported addition-type additives include dimethyl methylphosphonate (DMMP) [14,15,16], hexa-phenoxy-cyclotriphosphazene [17], polydopamine [18], ammonium polyphosphate [19,20], tris-(2-chloropropyl)-phosphate [21], polyhedral oligomeric silsesquioxane [22], dimethylpropanphosphonate [23], carbon nanotube [24], aluminum hydroxide (ATH) [25,26], magnesium hydroxide [27], expandable graphite (EG) [28], and so on. They all effectively enhance the flame retardancy of RPUFs. Additionally, reactive-type compounds have also been reported to endow RPUFs with excellent flame retardancy, such as [bis(2-hydroxyethyl)amino]-methyl-phosphonic acid dimethyl ester (BH) [29], phosphorylated soybean oil [30], phosphorylated polyols [31], etc. By means of these actions, RPUFs did not obtain high flame retardancy when they were utilized alone. As a consequence, the above flame retardants alone did not have a sufficient flame retardant efficiency.

Some of the flame retardant additives mainly quench free radical chain reactions in the gas phase; some of them mainly promote charring in the condensed phase; some of them simultaneously exert actions both in gas and condensed phases; and some of them react with other flame retardants to generate a better effect. However, when they are jointly employed in certain compositions, a higher flame retardant efficiency of RPUFs forms. In the reported literature, BH/EG systems in RPUFs can exert the addition of flame retardant effects [29] and DMMP/BH/EG systems in RPUFs generated continuously released flame retardant effects [32]. The two systems each brought a higher flame retardant efficiency to RPUFs than a single flame retardant additive.

According to the previous reports, the systems with a high flame retardant efficiency almost required the utilization of different flame retardant effects from different components.

In this thesis, the ternary system ATH/BH/EG was employed to construct high-performance flame retardant RPUFs. The ternary synergistic working mechanism of ATH/BH/EG on RPUFs was systematically investigated and discussed, which provided an effective way to construct novel synergistic flame retardant systems.

## 2. Experiment

### 2.1. Materials

(1) Polyether polyol (450L) was purchased from Dexin Lianbang Chemical Industry Co., Ltd. (Zibo, Shandong, China). The primary properties of DSU-450L were as follows: hydroxyl value, 450 ± 10 mg KOH equivalent/g; water content, ≤0.1 wt.%; viscosity (25 °C), 6000–10,000 mPa·s; potassium ion (K^+^), ≤8 mg/kg; pH, 4 to 6. (2) The 30% potassium acetate solution (KAc) was used as a catalyst and purchased from Liyang Yutian Chemical Co., Ltd. (Changzhou, Jiangsu, China). (3) Pentamethyldiethylenetriamine (Am-1), an effective catalyst for RPUFs, was obtained from Liyang Yutian Chemical Co., (Changzhou, Jiangsu, China). (4) *N*,*N*-Dimethylcyclohexylamine (DMCHA) was purchased from Jiangdu Dajiang Chemical Co., Ltd. (Yangzhou, Jiangsu, China). (5) The silicone foam stabilizer (SD-622) for RPUFs was purchased from Siltech New Materials Corporation (Suzhou, Jiangsu, China). (6) Deionized water was prepared in-laboratory, and was used as an auxiliary blowing agent. (7) 1,1-Dichloro-1-fluoroethane (HCFC-141b) was supplied by Hangzhou Fushite Chemical Industry Co., Ltd. (Hangzhou, Zhejiang, China), and was used as a blowing agent. (8) Polyphenylpolymethylene isocyanate (PAPI, 44V20) was purchased from German Bayer Company (Leverkusen, Germany). The primary performance indices were as follows: –NCO weight percent, 30%; monomer MDI content, 52%. (9) Expandable graphite (EG) (ADT 350) was produced by Shijiazhuang ADT Carbonic Material Factory (Shijiazhuang, Hebei, China). The primary properties of EG were as follows: moisture, 0.56%; pH, 7.0; expansion rate, 350 mL/g; volatility, 17.1%; ash, 4.8%; particle size (≥300 mm), 83%; and purity, ≥95%. (10) [Bis(2-hydroxyethyl)amino] methyl phosphonic acid dimethyl ester (BH) was supplied by Qingdao Lianmei Chemical Industry Co., Ltd. (Qingdao, Shandong, China). (11) Aluminum hydroxide (ATH) was supplied by Jinan Taixing Chemical Industry Co., Ltd. (Jinan, Shandong, China). (12) Aluminum oxide (AO) was purchased from Sinopharm Chemical Reagent Beijing Co., Ltd. (Beijing, China).

### 2.2. Preparation of RPUFs

ATH/BH/EG-filled RPUFs were prepared by box-foaming. The formulae are listed in Table 1. First of all, polyether polyol 450L, catalyst, 141b, and the three flame retardants ATH, BH, and EG were pre-mixed in a container using a stirrer to get a uniform mixture. Then, PAPI was immediately poured into the mixture, and the mixture was stirred at a high speed for 20 s. During expansion, the mixture was transferred into a mold (250 mm × 250 mm × 60 mm) to obtain a free-rise foam. After foaming, the samples were aged for 24 h. After aging, the foams were cut to the standard specimens. The sample without flame retardants was referred to as “neat RPUF”. All the flame retardant samples were named as follows: RPUF containing 14 wt.% ATH, 14 wt.% BH, and 6 wt.% EG was referred to as 14ATH/14B/6E/PU.

### 2.3. Characterization

The limited oxygen index (LOI) values were detected via Fire Testing Technology (Fire Testing Technology, London, UK) Dynisco LOI instrument according to ASTM D2863-97, and the sample dimensions were 100.0 mm × 10.0 mm × 10.0 mm.

The cone calorimeter test was characterized using an FTT instrument (Fire Testing Technology, London, UK) based on ISO5660 at an external heat flux of 50 kW/m^2^. The dimensions of the samples were 100.0 mm × 100.0 mm × 30.0 mm. The reported parameters were the average from two measurements.

The micro-morphology of the residual char from the cone calorimeter test with a conductive gold layer was observed using a scanning electron microscope (SEM, Tescan Vega II, Tescan SRO Co., Brno, Czech Republic) under high vacuum with a voltage of 20 kV.

The element compositions of the residues from cone calorimeter tests were investigated using the Perkin Elmer PHI 5300 ESCA X-ray photoelectron spectrometer (XPS) (Waltham, MA, USA). The selected residues were sufficiently grinded and mixed before analysis.

The apparent densities of the samples were calculated according to ISO 845:2006. The dimensions of the samples were 30.0 mm × 30.0 mm × 30.0 mm.

The compressive strength was tested using a CMT6004 electromechanical universal testing machine (MTS systems Co. Ltd., Shanghai, China) according to ISO 844-1787. The sample dimensions were 50.0 mm × 50.0 mm × 50.0 mm. The relative distortion of the compressed sample was more than 10%.

## 3. Results and Discussion

### 3.1. Flame Retardancy

To evaluate the preliminarily flame retardancy of RPUFs first, the LOI values of the specimens were investigated. The corresponding results are listed in Table 2. The incorporating fraction of ATH or AO inorganic components was 8 wt.% and 14 wt.%, whereas the fraction of BH/EG (mass ratio 14:6) was sustained unchanged at 20 wt.% in the RPUF matrix [29] because the ratio of BH/EG was the optimal one which could bring better flame retardancy to RPUFs in sifting formulae [29]. According to the LOI data in Table 2, the results showed that a 28 wt.% high fraction of ATH and AO increased the LOI values of RPUFs slightly when they were used alone. That said, when 8 wt.% ATH or AO were incorporated with 14 wt.% BH and 6 wt.% EG into RPUFs, their LOI values were sustained above 30%. If the fractions of AO or ATH in these systems were continuously increased to 14 wt.%, the LOI value of 14ATH/14B/6E/PU was further enhanced to 34%, whereas that of 14AO/14B/6E/PU did not obviously change. The results also revealed that both ATH and AO have the potential to impose flame retardancy to RPUFs, but ATH exerted the working effect more effectively than AO in the LOI test. Further, the LOI value of 8ATH/14B/6E/PU was nearly the same as that of the 19.6B/8.4E/PU system, but higher than 28ATH/PU, which preliminarily discloses that ATH/BH/EG has a ternary synergistic flame retardant effect on RPUFs.

In order to investigate the ternary synergistic reason of ATH/BH/EG systems’ efficacy in RPUF, and to explore the high-performance system in RPUFs, cone calorimeter and other tests were conducted, and the test results disclosed more clues. The typical data are listed in Table 2, including the peak value of heat release rate (PHRR), total heat release (THR), average effective heat of combustion (av-EHC), total smoke release (TSR), average yield of CO (av-COY), and average yield of CO_2_ (av-CO_2_Y). The curves of heat release rate (HRR) are represented in Figure 1, and mass loss curves are illustrated in Figure 2.

From Figure 1 and Table 2, the HRR value of neat RPUF dramatically increased and reached the maximum burning intensity after ignition, and the PHRR of neat RPUF reached 322 kW/m^2^, whereas the corresponding values of 28ATH/PU and 28AO/PU were respectively 215 and 285 kW/m^2^, indicating that ATH and AO alone can inhibit burning intensity, and that ATH has a stronger inhibition effect than AO. The ternary and binary flame retardant specimens, ATH/BH/EG, AO/BH/EG, and BH/EG, all obviously decreased the PHRR and THR values compared with neat RPUF. In contrast to BH/EG, ATH/BH/EG endowed RPUFs with a lower HRR value when the total addition fractions of flame retardants in RPUFs were 28 wt.%, whereas the AO/BH/EG samples did not show the same trend as ATH/BH/EG. 14ATH/14B/6E/PU also showed a lower HRR intensity than the comparison samples 19.6B/8.4E/PU and 14AO/14B/6E/PU before 200 s, which belonged to the main burning zone, indicating that three flame retardant components ATH/BH/EG synergistically inhibited burning intensity at a lower level. Moreover, an increased additional amount of ATH or AO in RPUF with constant 14 wt.% BH and 6 wt.% EG did not cause an obvious change in PHRR and THR. All the results obviously disclose that ATH has a better inhibition effect in burning intensity than AO, and that ATH can also exert a synergistic flame retardant effect in flame inhibition with BH/EG. This is probably caused by the endothermic decomposition and water release reaction from ATH.

Neither ATH nor AO brought an obvious reduction effect on THR in BH/EG/RPUF systems. ATH and AO added or not in RPUF with BH/EG did not obviously affect the av-EHC value, indicating that neither ATH nor AO had a quenching effect on the free radical chain reactions of combustion. Further, when they were evolved in PRUFs without other flame retardants, they caused stronger combustion.

According to TSR data, BH/EG systems did not suppress the smoke release because BH is a phosphorus-containing compound, which was decomposed to the fragments with a quenching effect to generate incomplete combustion. However, when ATH was incorporated into BH/EG/RPUFs, the TSR value of 14ATH/14B/6E/PU was reduced by 33.3% compared with that of 19.6B/8.4E/PU. The results implied that ATH had an outstanding smoke suppression effect in ATH/BH/EG systems. However, AO did not show similar effects on smoke release in AO/BH/EG systems. The only difference between ATH and AO is that ATH can undergo an endothermal reaction and decompose to release water before producing AO during combustion. The endothermal reaction will not directly suppress the smoke release, and neither will the product AO. The water from decomposed ATH should be determined as the only working component in suppressing smoke.

The normalized MLR curves from cone calorimeter are illustrated in Figure 2. When ATH or AO were incorporated into BH/EG/RPUFs, more decomposed fragments were reserved in residues and the mass loss rates were evidently reduced. In fact, AO does not decompose during combustion and it is be reserved in the residue directly. Thus, the mass loss ratio should be reduced. Although ATH has the capacity to decompose to release water and reduce mass, ATH still made more residue reserved in ATH/BH/EG/RPUFs than AO. The water from ATH should exert crucial actions during the charring process. It could be deduced that the phosphorus-containing fragments reacted with water to produce polyphosphate in the residue. Accordingly, the mass loss was reduced and the smoke release also was inhibited. In subsequent discussion, more evidence will be provided to supporting this deduction.

All the av-COY and av-CO_2_Y values of ATH/BH/EG/RPUFs or AO/BH/EG/RPUFs were decreased compared with neat RPUF. These results were similar to those of BH/EG/RPUFs. Therefore, the inhibition effect on flame from the ternary ATH/BH/EG system was a resulted of the BH/EG system.

### 3.2. The Analysis of Residual Char Digital Photos

Digital photos of the residual char of typical samples after the cone calorimeter test are shown in Figure 3. In Figure 3A, a little residue of the neat RPUF remained after burning, whereas 14AO/14B/6E/PU and 14ATH/14B/6E/PU obtained more residue than neat RPUF. It can be clearly observed that ATH and AO promoted the formation of a compact char layer. Compared with 14AO/14B/6E/PU, 14ATH/14B/6E/PU had a difference in char layer. A large amount of white powder appeared on the char surface of 14ATH/14B/6E/PU, which may have been caused by AO from decomposed ATH after dehydration. However, 14AO/14B/6E/PU directly containing AO did not show the same phenomenon. This is because AO powders were covered by residue fragments produced by the matrix. The results also testified that the water from ATH did not directly react with the smoke fragments in the gas phase, and that it should capture the phosphorus-containing components in the condensed phase, and thus less smoke was formed. The working effect of water from ATH resulted in white AO powder from decomposed ATH appearing on the residue surface.

### 3.3. SEM and Elemental Analysis of Residues after Cone Calorimeter Test

In order to further disclose the flame retardant mechanism of ATH/BH/EG systems, SEM photos of residue were obtained from 14ATH/14B/6E/PU and 14AO/14B/6E/PU samples, and they are listed in Figure 4. In Figure 4A,a, the SEM photos from 14ATH/14B/6E/PU clearly show that the tiny solid particles were evenly dispersed and adhered to the residue surface. The combination of AO powders from decomposed ATH with residue implied that the deduced reaction between them had occurred. The reason is that the produced AO particles from ATH will be packed in residue if there is no reaction between ATH and the decomposed matrix, as AO and the matrix in Figure 4B,b show. In the 14AO/14B/6E/PU residue sample, many aggregated globular AO particles were wrapped up by residue, which made AO not fully covered on the char surface to form a protective effect. Therefore, as the previous deduction concludes, the water released from ATH should directly react with phosphorus-containing components in the matrix before the components are released to the gas phase. ATH with BH/EG fully jointly interacted to lock in more components from the matrix in the residue, which is why ATH/BH/EG generated ternary synergistic flame retardant effects in the condensed phase.

To further investigate the mechanism of ATH’s action in 14ATH/14B/6E/PU, the elemental ratios in residues were detected using XPS. Table 3 lists three residue samples after the cone calorimeter test. Compared with 14AO/14B/6E/PU and 19.6B/8.4E/PU, the residue of 14ATH/14B/6E/PU obviously had a greater phosphorus content, verifying that ATH exerted the capturing action on phosphorus-containing components during combustion. As per the deduction in the previous discussion, the water released from ATH should react with the decomposition products from an aliphatic phosphorus-oxide structure to generate polyphosphoric acid, and then form polyphosphate, thereby increasing the barrier effect of the char layer to heat and flame. In order to confirm this deduction, we further analyzed the energy spectrum of the phosphorus and aluminum to seek more direct evidence.

The different phosphorus binding energies from XPS in Figure 5 support the previous deduction. The binding energy values 131.5 and 131.0 eV of phosphorus in 14AO/14B/6E/PU and 19.6B/8.4E/PU residues were close to each other, implying that the phosphorus elements stayed in similar chemical structure, and AO just slightly interacted with phosphorus-containing components from BH. However, the phosphorus in the residue of 14ATH/14B/6E/PU showed a different binding energy result. The binding energy value of the phosphorus raised to 132.5 eV, which is closer to the 132.8 eV of phosphorus in melamine polyphosphate (MPP). This result reveals that the phosphorus content from BH in 14ATH/14B/6E/PU was converted to a polyphosphate structure, as is the phosphorus style in MPP. Since the phosphorus components in 14AO/14B/6E/PU and 19.6B/8.4E/PU all did not generate similar binding energy values, it can be determined that the existence of ATH helped the phosphorous components to form a polyphosphate structure. Further, the one difference between ATH and AO in decomposition products is that ATH can release water during decomposition. Water from ATH should be the crucial medium to help phosphorus in BH to transform to a polyphosphate structure.

### 3.4. Ternary Synergistic Flame Retardant Mechanism of ATH/BH/EG in RPUFs

In ATH/BH/EG ternary flame retardant systems, the condensed-phase synergistic flame retardant mechanism of ATH with BH/EG was systematically investigated. The ATH/BH/EG ternary synergistic flame retardant mechanism is illustrated in Figure 6. When the RPUFs containing ATH/BH/EG were ignited, EG expanded rapidly and formed a worm-like thermal-insulating layer. At the same time, water molecules were produced from decomposed ATH. Then, the water directly combined or captured phosphate-containing components or their derivatives to generate polyphosphate. The worm-like expanded graphite, AO, and polyphosphate combined with each other to form a phosphorus-rich compact char layer. The char layer not only prevented the release of combustible gas, but also inhibited the thermal feedback to the RPUF matrix and further effectively decreased the decomposition velocity of the matrix. The formation of polyphosphate also locked more phosphorus and carbonaceous components in the residue, which reduced the “fuel” and smoke density. ATH/BH/EG generated a well-working condensed-phase ternary synergistic effect, which can endow RPUFs with better flame retardant performance.

### 3.5. Physical Properties

As a thermal insulation material, RPUFs must meet the demand of flame retardant performance, and at the same time also need to possess the necessary physical-mechanical properties. The physical properties of the samples (i.e., compressive strength and apparent density) were tested, and the results are listed in Table 4. Apparent density is a very important factor in the usability of RPUFs. ATH, AO, and EG are solid fillers with a higher density in flame retardant systems. Therefore, with an increasing ratio of AO and ATH, the apparent density of RPUFs also increased. From Table 4, the incorporation of AO and ATH led to increasing apparent density of RPUFs to between 47 and 56 kg·m^−3^. The apparent density can be accepted as a measure of heat insulation flame retardant RPUF materials in practice. The data of apparent density also indicate that the ATH/BH/EG flame-retardant system did not affect the foaming process. Compression strength can reveal the mechanical properties and usability of the RPUF materials. Through the previous discussion, ATH/BH/EG systems can endow better flame retardant performance than AO/BH/EG systems, but AO/BH/EG systems have a higher compression strength.

## 4. Conclusions

In this work, the ternary synergistic flame retardant behavior of ATH/BH/EG in RPUFs was investigated. Compared with AO/BH/EG and BH/EG flame retardant systems, ATH/BH/EG with the same fractions obviously increased the LOI value, decreased the PHRR, and effectively inhibited the TSR of RPUFs. The better flame retardancy of ATH/BH/EG systems in RPUFs was a result of the condensed-phase ternary synergistic flame retardant effect. During combustion, the water molecules from decomposed ATH directly reacted with or captured phosphorus-containing components to generate polyphosphate, reserved in a char layer. The worm-like expanded graphite, AO from decomposed ATH, and polyphosphate combined with each other to form a char layer with a better barrier effect to endow RPUFs with better flame retardancy. ATH, BH, and EG in RPUF can jointly work together and generate condensed-phase synergistic flame retardant effects during combustion.

## Figures and Tables

**Figure 1 polymers-11-00207-f001:**
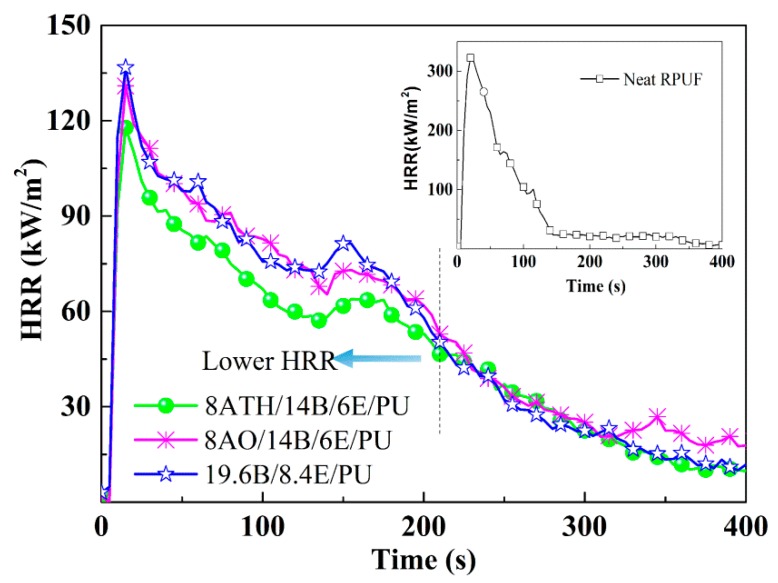
Heat release rate (HRR) curves of typical flame retardant rigid polyurethane foams (RPUFs).

**Figure 2 polymers-11-00207-f002:**
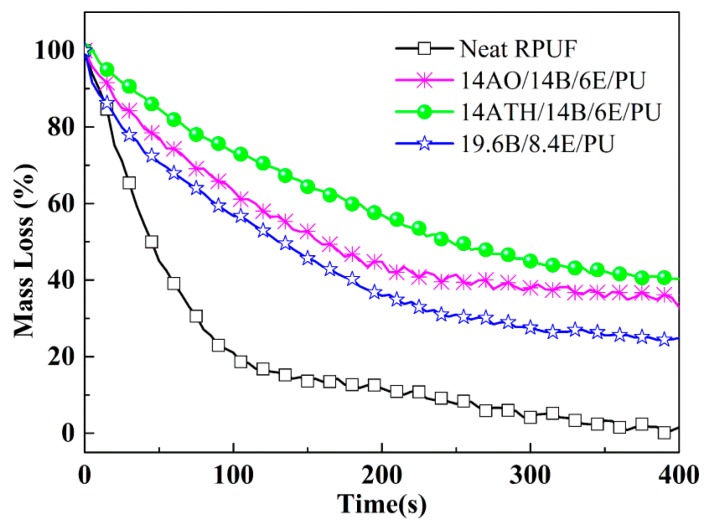
Mass loss curves of RPUFs.

**Figure 3 polymers-11-00207-f003:**
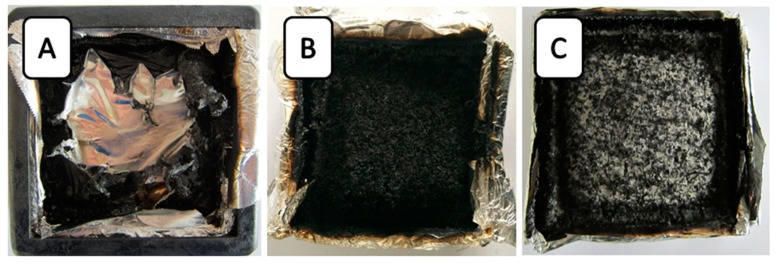
Digital photos of residues from the cone calorimeter test. (**A**) Neat RPUF; (**B**) 14AO/14B/6E/PU; (**C**) 14ATH/14B/6E/PU.

**Figure 4 polymers-11-00207-f004:**
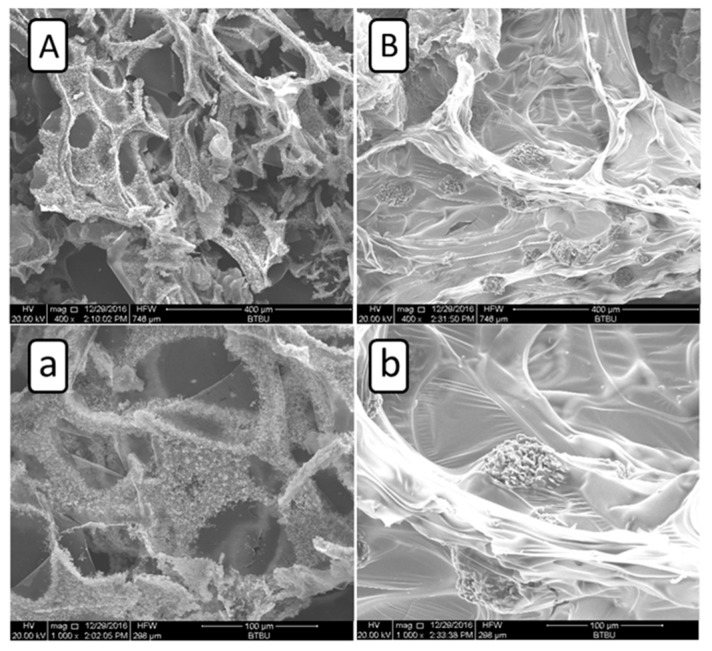
SEM photos of residues after the cone calorimeter test. (**A**) 400×, 14ATH/14B/6E/PU; (**a**) 1000×, 14ATH/14B/6E/PU; (**B**) 400×, 14AO/14B/6E/PU; (**b**) 1000×, 14AO/14B/6E/PU.

**Figure 5 polymers-11-00207-f005:**
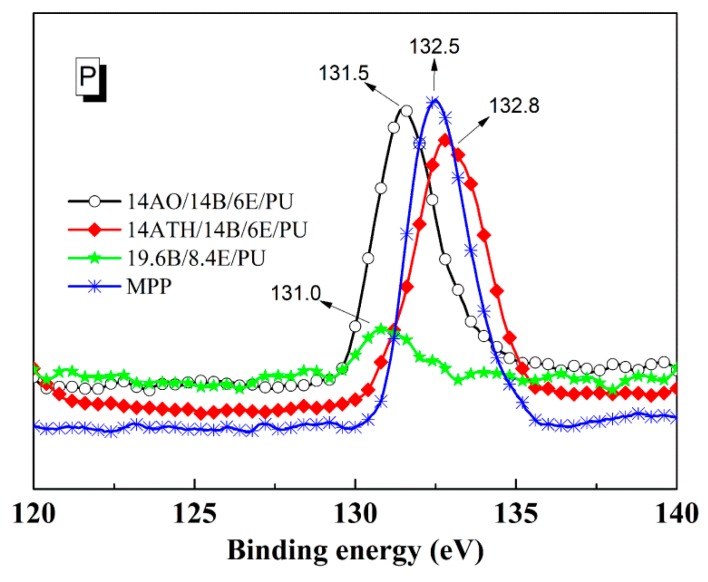
Energy spectra of phosphorus in residues and melamine polyphosphate (MPP).

**Figure 6 polymers-11-00207-f006:**
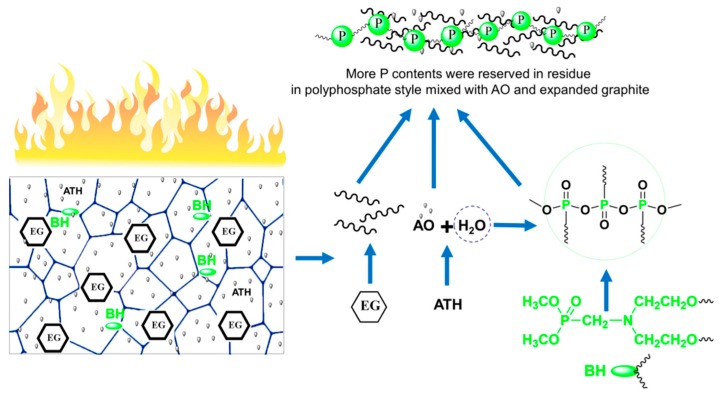
Flame retardant mechanism of ATH/BH/EG system in RPUFs.

**Table 1 polymers-11-00207-t001:** Formulae of flame retardant rigid polyurethane foams (RPUFs) *^a^*.

Samples	FR Ratio (%)	ATH (g)	AO (g)	BH (g)	EG (g)	450L (g)
Neat RPUF	0%	--	--	--	--	72.0
8ATH/14B/6E/PU	8%ATH/14%BH/6%EG	19.1	--	33.5	14.3	43.0
14ATH/14B/6E/PU	14%ATH/14%BH/6%EG	35.5	--	35.5	15.5	43.0
8AO/14B/6E/PU	8%ATH/14%BH/6%EG	--	19.1	33.5	14.3	43.0
14AO/14B/6E/PU	14%ATH/14%BH/6%EG	--	35.5	35.5	15.5	43.0
19.6B/8.4E/PU	BH:EG 14:6	--	--	46.3	20.1	43.0
28ATH/PU	28%ATH	78.0	--	--	--	72.0
28AO/PU	28%AO	--	78.0	--	--	72.0

*^a^* Catalyst 6.2 g, 141b 14.4 g, and polyphenylpolymethylene isocyanate (PAPI) 108 g in every formula. 450L: polyether polyol; ATH: aluminum hydroxide; AO: aluminum oxide; BH: [bis(2-hydroxyethyl)amino]-methyl-phosphonic acid dimethyl ester; EG: expandable graphene.

**Table 2 polymers-11-00207-t002:** Tested results of LOI and cone calorimeter test (0–400 s).

Samples	LOI (%)	PHRR (kW/m^2^)	av-EHC (MJ/kg)	THR (MJ/m^2^)	TSR (m^2^/m^2^)	av-COY (kg/kg)	av-CO_2_Y (kg/kg)
Neat RPUF	19.4	322 ± 8	20.8 ± 1.0	27.1 ± 0.8	899 ± 22	0.24 ± 0.04	2.52 ± 0.23
8ATH/14B/6E/PU	31.2	120 ± 2	18.5 ± 0.0	19.4 ± 0.3	625 ± 19	0.20 ± 0.02	2.20 ± 0.14
14ATH/14B/6E/PU	34.0	117 ± 2	18.8 ± 0.8	20.1 ± 0.1	496 ± 16	0.21 ± 0.05	2.31 ± 0.04
8AO/14B/6E/PU	30.6	129 ± 2	17.1 ± 0.7	21.7 ± 0.4	709 ± 17	0.21 ± 0.03	1.95 ± 0.30
14AO/14B/6E/PU	30.7	129 ± 2	17.3 ± 0.4	21.1 ± 0.7	707 ± 28	0.28 ± 0.00	2.48 ± 0.02
19.6B/8.4E/PU	31.0	132 ± 5	18.2 ± 0.4	20.9 ± 0.7	744 ± 21	0.22 ± 0.01	2.36 ± 0.01
28ATH/PU	22.3	285 ± 15	21.0 ± 0.1	35.5 ± 0.3	1165 ± 34	0.19 ± 0.03	2.32 ± 0.50
28AO/PU	24.5	215 ± 10	18.5 ± 0.8	32.4 ± 0.5	1091 ± 47	0.18 ± 0.01	2.64 ± 0.07

**Table 3 polymers-11-00207-t003:** Elemental contents of residues from flame retardant RPUFs.

Samples	C (%)	N (%)	O (%)	P (%)	Al (%)
14ATH/14B/6E/PU	55.33	3.69	26.18	9.37	5.41
14AO/14B/6E/PU	66.21	5.64	17.45	5.62	5.34
19.6B/8.4E/PU	81.86	6.31	8.84	2.28	0

**Table 4 polymers-11-00207-t004:** Apparent density and compression strength of RPUFs.

Samples	Apparent Density (kg/m^3^)	Compression Strength (MPa)
Neat RPUF	35.2	0.20
8ATH/14B/6E/PU	47.4	0.19
14ATH/14B/6E/PU	55.5	0.18
8AO/14B/6E/PU	51.8	0.24
14AO/14B/6E/PU	54.8	0.27
19.6B/8.4E/PU	49.6	0.22

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
