# Peer review of "Flame Retardant Behavior of Ternary Synergistic Systems in Rigid Polyurethane Foams"

_polymers, 2019, doi:10.3390/polym11020207_

Round 1

Reviewer 1 Report

This is a paper on fire retardance of rigid Polyurethane foams. The synergistic effect of introduction of Aluminum oxide (AO) or Aluminum hydroxide (ATH) in a phosphonate/expanded Graphite fire retardant formulation is presented and discussed in term of Limit Oxygen Index, Cone calorimeter parameters and various techniques of characterization. The paper is well organized and nicely to be read.

In summary, it has been shown that Aluminum hydroxide performs better as a synergist because of dehydration water evolved during heating and combustion of the samples. However, I think that some points in the discussion of the mechanism have to be clarified.

1)      Line 161-164: “ATH and AO all did not bring an obvious reduction effect on THR in BH/EG/RPUF system. It is  attributed to the increased matrix density and the amounts of combustible materials in same volume  after adding solid ATH and AO. More “fuels” led to higher heat release”
- why increased matrix density could induce reduction of THR? 
- I do not understand if fuel volume is increased in the presence of AO or ATH or not, Please clarified what is the role of the amount on fuel in THR

2)      Line 176-177 “The normalized MLR curves from cone calorimeter are illustrated in Figure 2. When ATH or AO  were incorporated into BH/EG/RPUFs, more decomposed fragments were reserved in residues and  the mass loss rates were evidently reduced.
- the difference in mass loss between Aluminum containing mixtures ant the reference FR mixture correspond nearly to the content of inorganic component (nearly 14%) . So, in my view, this is not a demonstration that more decomposed fragment were reserved in the residue. Please clarify and reformulate

3)       The authors stated that the evolution of water from ATH lead to the formation of Polyphosphates structure. The only proof is the XPS spectra that I’m not sure is enough selective to definitely attribute  the P structure variation. Why do not consider hydrolysis of phosphonate, phosphonic acid formation and eventually absorption of phosponate on aluminum oxide surface?

Author Response

This is a paper on fire retardance of rigid Polyurethane foams. The synergistic effect of introduction of Aluminum oxide (AO) or Aluminum hydroxide (ATH) in a phosphonate/expanded Graphite fire retardant formulation is presented and discussed in term of Limit Oxygen Index, Cone calorimeter parameters and various techniques of characterization. The paper is well organized and nicely to be read.

In summary, it has been shown that Aluminum hydroxide performs better as a synergist because of dehydration water evolved during heating and combustion of the samples. However, I think that some points in the discussion of the mechanism have to be clarified.

1. Line 161-164: “ATH and AO all did not bring an obvious reduction effect on THR in BH/EG/RPUF system. It is  attributed to the increased matrix density and the amounts of combustible materials in same volume  after adding solid ATH and AO. More “fuels” led to higher heat release” - why increased matrix density could induce reduction of THR? 
- I do not understand if fuel volume is increased in the presence of AO or ATH or not, Please clarified what is the role of the amount on fuel in THR.

Response: Thank you for reviewer’s suggestion. The discussion on fuels was deleted. According to the results, when ATH and AO were evolved in PRUFs without other flame retardants, they caused stronger combustion. So, we can understand why ATH and AO can not reduce the THR values.

2. Line 176-177 “The normalized MLR curves from cone calorimeter are illustrated in Figure 2. When ATH or AO  were incorporated into BH/EG/RPUFs, more decomposed fragments were reserved in residues and  the mass loss rates were evidently reduced.
- the difference in mass loss between Aluminum containing mixtures ant the reference FR mixture correspond nearly to the content of inorganic component (nearly 14%) . So, in my view, this is not a demonstration that more decomposed fragment were reserved in the residue. Please clarify and reformulate

Response: The reduced fragments were not only proved by the increased residues after combustion, but also testified by the reduced smoke release. The fragments such as smoke particles were absorbed in char layer. And ATH has better effect on absorbing smoke particles, which should be pyrolysis fragments.

3. The authors stated that the evolution of water from ATH lead to the formation of Polyphosphates structure. The only proof is the XPS spectra that I’m not sure is enough selective to definitely attribute  the P structure variation. Why do not consider hydrolysis of phosphonate, phosphonic acid formation and eventually absorption of phosponate on aluminum oxide surface?

Response: We agree the reviewer’s opinion. We just give the final state of phosphorus contents--polyphosphate because the phosphonate and phosphonic acid would form polyphosphate after combustion. So, we did not discuss the middle-state of phosphorus contents because the evidence is difficult to grasp.

Reviewer 2 Report

Recipes of foams should be presented per hundred parts/grams of polyol, it is much more clear this way. Also Authors should mention isocyanate index used, which is crucial for the performance of rigid PU foams. 

What were the density values of prepared foams? Authors have to present them, since apparent density determines almost all properties of foamed materials. Authors obviously know that, because they are writing about the influence of density on THR. So, apparent density needs to be presented. 

In the description of LOI and cone calorimeter results Authors should refer more to the mechanisms of flame retardancy of all applied compounds, not only AO and ATH. Maybe some chemical reactions would make discission more advanced. I know that description is presented at the end, but manuscript and description of results would be better when mechanisms will be directly linked to the particular results, e.g. THR, TSR, HRR etc.

Author Response

1.Recipes of foams should be presented per hundred parts/grams of polyol, it is much more clear this way. Also Authors should mention isocyanate index used, which is crucial for the performance of rigid PU foams. 

Response: The per hundred parts/grams of polyol is correct usage in reality. But in this research, we discussed the flame retardantcy of systems in same mass ratio of different flame retardants.

2. What were the density values of prepared foams? Authors have to present them, since apparent density determines almost all properties of foamed materials. Authors obviously know that, because they are writing about the influence of density on THR. So, apparent density needs to be presented. 

Response: The apparent density has been replenished and discussed.

3. In the description of LOI and cone calorimeter results Authors should refer more to the mechanisms of flame retardancy of all applied compounds, not only AO and ATH. Maybe some chemical reactions would make discission more advanced. I know that description is presented at the end, but manuscript and description of results would be better when mechanisms will be directly linked to the particular results, e.g. THR, TSR, HRR etc.

Response: The flame retardant mechanism of BH and EG have been researched in the previous literature by us. This research focused on the flame retardant behavior after incorporating ATH and AO to BH/EG system. So, the main discussion is that how ATH and AO affected or co-worked with BH/EG.